# Breaking the Glass Ceiling for Embedding-Based Classifiers for Large Output Spaces

**Chuan Guo**[*][†]
Cornell University
cg563@cornell.edu

**Ali Mousavi**[*]
Google Research
alimous@google.com

**Xiang Wu**[†]
ByteDance
xiang.wu@bytedance.com

**Daniel Holtmann-Rice**
Google Research
dhr@google.com

**Satyen Kale**
Google Research
satyenkale@google.com

**Sashank Reddi**
Google Research
sashank@google.com

**Sanjiv Kumar**
Google Research
sanjivk@google.com

## Abstract

In extreme classification settings, embedding-based neural network models are currently not competitive with sparse linear and tree-based methods in terms of accuracy. Most prior works attribute this poor performance to the low-dimensional bottleneck in embedding-based methods. In this paper, we demonstrate that theoretically there is no limitation to using low-dimensional embedding-based methods, and provide experimental evidence that overfitting is the root cause of the poor performance of embedding-based methods. These findings motivate us to investigate novel data augmentation and regularization techniques to mitigate overfitting. To this end, we propose GLaS, a new regularizer for embedding-based neural network approaches. It is a natural generalization from the graph Laplacian and spread-out regularizers, and empirically it addresses the drawback of each regularizer alone when applied to the extreme classification setup. With the proposed techniques, we attain or improve upon the state-of-the-art on most widely tested public extreme classification datasets with hundreds of thousands of labels.

## 1   Introduction

We study the problem of multi-label classification with large output space, which has garnered significant attention in recent years [36, 6, 14, 3, 33, 23]. This problem differs from the traditional classification setting insofar that the number of labels is potentially in the millions, presenting significant computational challenges. Many real world applications such as product recommendation and text retrieval can be formulated under this framework and thus, practical solutions to this problem can have significant and far-reaching impact.

In this unusual yet practical setting, both the number of input feature dimensions $D$ and the number of labels $K$ could be upwards of hundreds of thousands or even millions. This renders most traditional machine learning models, such as logistic regression and SVM, infeasible due to excessive number of model parameters — approximately $O(DK)$. Most recent approaches resort to using sparse linear

---

[*]Equal Contribution
[†]Work done at Google

models or tree-based methods in order to tackle this challenge [29, 23, 24, 34, 33]. An alternate approach to address this problem is through low-dimensional embeddings. Here, the model consists of an *embedding function* $\phi : \mathbb{R}^D \to \mathbb{R}^d$, where $d$ is the *embedding dimension*, and a *classifier* $f : \mathbb{R}^d \to \{0, 1\}^K$. Thus, for any input $\mathbf{x} \in \mathbb{R}^D$, $f(\phi(\mathbf{x}))$ is the indicator vector or *label vector* of the predicted labels. To handle a large number of labels, the embedding dimension $d$ is chosen to be small in comparison to $D$; thereby, significantly reducing the number of model parameters.

Despite their accomplishments in computer vision and natural language processing domains [17, 27], embedding-based deep neural networks (DNNs) have *not* achieved the same level of success in learning with large output spaces. This point is often attributed to low-dimensional bottleneck layers in neural networks that cannot represent enough information for the downstream learning task when the number of potential labels is substantially larger than the embedding dimensionality [24, 6, 32]. Attempts to circumvent this limitation have been met with limited success [6, 31]. As a result, sparse linear models and tree-based methods are favored in comparison to embedding-based methods for large-scale multi-label classification problems.

In this paper, we investigate embedding-based methods for the problem of our interest. Our main observation is that, contrary to the widespread belief of limited representation power, overfitting is the cause for the inferior performance of embedding-based methods, which suggests that efforts to either augment the training set or regularize the model may dramatically boost test set performance. Inspired by this, we show that a number of regularization techniques can shrink the generalization gap for embedding-based methods and allow them to achieve, or improve upon, state-of-the-art accuracy on a variety of widely tested public datasets. The most discernible improvement comes from a novel regularizer that promotes embeddings for frequently co-occurring labels to be close.

**Contributions.** In the light of this background, we state the following key contributions of this paper:

1. We demonstrate experimentally that the main reason for the poor performance of neural network embedding-based models is *overfitting*. Our empirical observation is further supported by theoretical analysis, where we prove that there exists a low-dimensional embedding-based linear classifier with perfect accuracy in the limit of infinite expressivity of the embedding map. This shows that, contrary to speculations in existing literature, low-dimensional embeddings are indeed sufficiently expressive and cannot be a bottleneck.

2. Based on this finding, we propose a suite of principled data augmentation and regularization techniques, including a novel regularizer called GLaS, to shrink the gap between training and test performance.

3. Finally, on several widely tested public datasets, with our proposed techniques, we achieve state-of-the-art results with very simple network architectures and little tuning. We achieve high precision and propensity scores, thus demonstrating the effectiveness of our method even on infrequent tail labels. We also provide an ablation study to highlight the effectiveness of each individual factor. This provides a strong baseline and several new venues for future research on applying embedding-based methods to the large output space setting.

## 1.1 Related Work

There is a vast amount of literature on text classification; therefore, we only mention those that are most relevant to the problem setting of our interest. Existing approaches to our problem setting can be broadly classified into three categories: (i) Embedding-based methods, (ii) Tree-based methods and (iii) Sparse and One-vs-all methods. We discuss these approaches briefly here.

**Embedding-based methods** learn a model of the form $f(\phi(\mathbf{x}))$ where $\phi(\mathbf{x}) \in \mathbb{R}^d$ and $d$ is small. Embedding methods mainly differ in their choice of the functional form and approaches to learn the parameters of the function. A variety of approaches such as compressed sensing [12], bloom filter [10], and SVD [36] are applied to train these models. While most of these approaches assume a linear functional form [7, 9, 18, 28], non-linear forms have also been proposed [6]. One criticism of embedding-based approaches is that label embeddings are compressed to a very small dimensionality $d$, which is believed to cause degradation in performance greatly [24, 6] and are thus, less favored for large-scale settings.

**Tree-based methods** learn a hierarchical structure over the label space and predict the path from the root to the target label [1, 15, 29, 24, 14, 22, 35]. While this greatly reduces inference time and the

number of parameters needed to be learnt, it typically comes at the cost of low prediction accuracy. Although traditionally done over the label set [29], more recent methods [24, 14] partition the feature space instead, relying on the assumption that only a small set of features are relevant for any label. These methods are heavily affected by so-called *cascading* effect, where the prediction error at the top cannot be corrected at a lower level.

**Sparse and One-vs-all methods** restrict the model capacity and improve efficiency by applying *sparse linear* methods to learn only a small fraction of the non-zero parameters. This allows the sparse model to be kept in main memory while ensuring that matrix-vector products can be carried out efficiently. Methods such as DiSMEC [3], ProXML [4], PD-Sparse [34] and PPD-Sparse [33] are representative of this strategy and have enjoyed great success recently. DiSMEC and PPD-Sparse are, in particular, highly parallelizable since they are based on the one-vs-all approach for training extreme multi-label classification models. However, these models are typically simple linear models and hence, do not capture complex non-linear relationships.

## 2 Discussion on Embedding-based Methods

In this section, we describe our problem setup more formally and investigate the validity of the criticism on embedding-based methods. The general learning problem of multi-label classification can be defined as follows. Given an input $\mathbf{x} \in \mathcal{X} \subset \mathbb{R}^D$, its label $\mathbf{y} \in \mathcal{Y} \subset \{0, 1\}^K$ is a $K$-dimensional vector with multiple non-zero entries, where $\mathbf{y}^{(k)} = 1$ if and only if label $k$ is relevant for input $\mathbf{x}$. Let $L_{\mathbf{y}}$ denote the set of indices that are non-zero in $\mathbf{y}$. The elements of the set $L_{\mathbf{y}}$ are, hereafter, referred to as *relevant labels* in $\mathbf{y}$. The number of distinct labels $K$ is assumed to be large (on the order of hundreds of thousands or even millions). The goal of all embedding-based methods is to learn a model of the form $f(\phi(\mathbf{x})) : \mathcal{X} \to \{0, 1\}^K$ where $\phi(\mathbf{x}) \in \mathbb{R}^d$ and $d \ll D, K$ and $f : \mathbb{R}^d \to \{0, 1\}^K$ is a classifier on top of the embedding.

The most common form of $f$ is a linear classifier. A linear classifier is parameterized by a *label embedding matrix* $\mathbf{V} \in \mathbb{R}^{d \times K}$ which is used to predict *scores* for all labels by computing $\phi(\mathbf{x})^\top \mathbf{V}$. $\mathbf{V}$ is called a label embedding matrix since its columns can be interpreted as embeddings of the $K$ labels in the same embedding space, $\mathbb{R}^d$. In the following, for a label $y$, we will use the notation $\mathbf{v}_y$ to denote the embedding of $y$ given by $\mathbf{V}$, i.e the $y$-th column of $\mathbf{V}$. Depending on the specific formulation, the set of labels predicted for the input $\mathbf{x}$ can then be obtained by thresholding the scores at some value $\tau$, i.e., $\{y : \phi(\mathbf{x})^\top \mathbf{v}_y \geq \tau\}$ or taking the top $m$ largest scores, i.e., $\text{Top}(\phi(\mathbf{x})^\top \mathbf{V}, m)$.

The use of a linear classifier on top of embeddings naturally leads to a low-rank structure for the score vectors of the labels: the set $\{\phi(\mathbf{x})^\top \mathbf{V} : \mathbf{x} \in \mathcal{X}\}$ has rank at most $d$. This restriction on the score vectors has frequently been cited as a reason for the poor performance of embedding based approaches for extreme classification problems. However, several studies [31, 6] show that the set of *label* vectors violates the low-rank structure on large-scale datasets. We should note that the label vectors are generated by either thresholding the scores or taking the top $m$ highest scores, which is a highly non-linear transformation. Thus, it is not immediately clear if the low-rank structure of the score vectors directly translates to a low-rank structure on the label vectors.

There have been efforts to tackle this presumed issue of embedding-based methods, primarily by using a more complex final classifier $f$ than simple linear ones. For instance, Xu et al. [31] decomposed the label matrix into a low-rank and a sparse part, where the sparse part captures tail labels as outliers. Bhatia et al. [6] developed an ensemble of local distance preserving embeddings to predict tail labels. In particular, they cluster data points into sub-regions and use a $k$-nearest neighbor classifier in the locally learned embedding space. However, these modern embedding-based approaches have several drawbacks [3] and cannot outperform other approaches on all large-scale datasets.

While most sparse linear and tree-based methods outperform embedding-based approaches, there has not been any definitive proof that the inherent problem with embedding-based methods is their use of low-dimensional representations for the score vectors. To the contrary, we provide experimental evidence that a low-dimensional embedding produced by training a simple neural network extractor can attain near-perfect training accuracy but generalize poorly, suggesting that overfitting is the root cause of the poor performance of embedding-based methods that has been reported in the literature. In fact, we will show that theoretically there is no limitation to using low-dimensional embedding-based methods, even with simple linear classifiers.

## 2.1 Validity of Low-Dimensional Bottleneck Criticism

We first present a different perspective regarding embedding-based models, showing their inferior performance in large output spaces is due to overfitting to training set rather than their inability to represent the input-label relationship with low-dimensional label embeddings.

Let $\phi_{\mathbf{w}}$ be the embedding function parameterized by some vector $\mathbf{w}$ that takes as input $\mathbf{x} \in \mathcal{X}$ and outputs a feature embedding $\phi_{\mathbf{w}}(\mathbf{x}) \in \mathbb{R}^d$. In practice, we may take $\phi_{\mathbf{w}}$ to be a linear function $\phi_{\mathbf{w}}(\mathbf{x}) = \mathbf{w}^\top \mathbf{x}$ or a neural network with multiple linear layers and ReLU activation. We use a linear classifier on top of the embedding, parameterized by a matrix $\mathbf{V} \in \mathbb{R}^{d \times K}$, whose columns give the label embeddings $\mathbf{v}_y$ for all labels $y$. Define the scoring function $h : \mathcal{X} \to \mathbb{R}^K$ as $h(\mathbf{x}) = \phi_{\mathbf{w}}(\mathbf{x})^\top \mathbf{V}$. At training time, we sample an input-label pair $(\mathbf{x}, \mathbf{y})$ uniformly and compute the margin loss [20]:

$$\ell(h(\mathbf{x}), \mathbf{y}) := \sum_{y \in L_{\mathbf{y}}} \sum_{y' \notin L_{\mathbf{y}}} [h(\mathbf{x})_{y'} - h(\mathbf{x})_y + c]_+ \tag{1}$$

Recall that $L_{\mathbf{y}}$ denotes the set of indices that are non-zero in $\mathbf{y}$. This loss encourages the scores for all relevant labels to be higher than the scores for irrelevant labels by a margin of $c > 0$. However, since the set of labels is large, computing this sum over the entire set is prohibitively expensive during training. Instead, we use a *stochastic estimate* of the loss by sampling a small subset of labels from $L_{\mathbf{y}}$ and computing the sum over that subset only. This loss function can be efficiently minimized using batched stochastic gradient descent. An alternative option is to use the so-called stochastic negative mining loss [25]. Algorithm 1 summarizes the training procedure.

We now illustrate the overfitting issue on this embedding-based model setup. Figure 1 shows the results of training our model on the AMAZONCAT-13K dataset. The statistics of this dataset is summarized in Table 5 in the

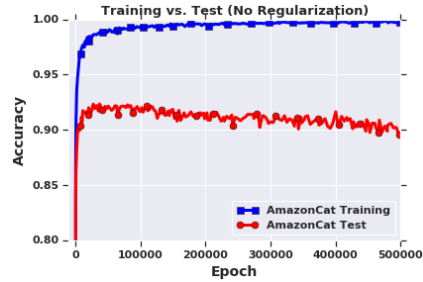

Figure 1: Training (blue) and test (red) accuracy of Alg. 1 on the AMAZONCAT-13K dataset. The non-regularized embedding-based method severely overfits to the training data.

supplementary material. The blue line shows that training accuracy continues to improve throughout optimization, culminating in near-perfect accuracy towards the end of training. We emphasize that this disputes the argument made by previous works that embedding-based models are ill-suited for this dataset due to the dimensionality constraint. However, we observe in Figure 1 is that our embedding-based model has severely overfitted to the training set. This observation highlights the need for regularization techniques to improve the performance of embedding-based methods.

---

**Algorithm 1** Training the basic embedding model

---

1: **Input**: Dataset $\{(\mathbf{x}_1, \mathbf{y}_1), \ldots, (\mathbf{x}_n, \mathbf{y}_n)\}$
2:        Feature embedding model $\phi_{\mathbf{w}} : \mathcal{X} \to \mathbb{R}^d$
3:        Label embedding matrix $\mathbf{V} \in \mathbb{R}^{d \times K}$
4:        Loss function $\ell : \mathbb{R}^K \times [K] \to \mathbb{R}$
5:        Learning rates $\eta_{\mathbf{w}}, \eta_{\mathbf{V}}$
6: Initialize $\mathbf{w}, \mathbf{V}$
7: **repeat**
8:     Sample a batch $\mathbf{x}_1, \ldots, \mathbf{x}_B$
9:     Sample indices $k_1, \ldots, k_B$ uniformly from non-zero indices of $\mathbf{y}_1, \ldots, \mathbf{y}_B$
10:     Compute loss $L \leftarrow \frac{1}{B} \sum_{i=1}^{B} \ell(\phi_{\mathbf{w}}(\mathbf{x}_i)^\top \mathbf{V}, k_i)$
11:     Compute gradients $\frac{dL}{d\mathbf{w}}$ and $\frac{dL}{d\mathbf{V}}$ via backpropagation
12:     Update $\mathbf{w} \leftarrow \mathbf{w} - \eta_{\mathbf{w}} \frac{dL}{d\mathbf{w}}, \mathbf{V} \leftarrow \mathbf{V} - \eta_{\mathbf{V}} \frac{dL}{d\mathbf{V}}$
13: **until** convergence

---

## 2.2 Existence of Perfect Accuracy Low-Dimensional Embedding Classifiers

We further support our argument theoretically and demonstrate the fact that embedding-based models can attain near-perfect accuracy is not limited to any specific dataset, but is feasible in general. We

make the following mild assumption on the data: for every $\mathbf{x}$ there exists a unique label vector $\mathbf{y} = \mathbf{y}(\mathbf{x})$, and the number of non-zero entries in $\mathbf{y}(\mathbf{x})$ is bounded by $s \ll K$, i.e., the number of true labels associated with any feature vector is at most some small constant $s$. Under this assumption, the following result shows that low-dimensional embedding-based models do not suffer from inability to represent the input-label relationship. Proof can be found in the supplementary material.

**Theorem 2.1.** *Let $\mathcal{S} \subseteq \mathcal{X}$ be a sample set. Under the assumption on the data specified above, there exists a function $\phi : \mathcal{X} \to \mathbb{R}^d$, and a label embedding matrix $\mathbf{V} \in \mathbb{R}^{d \times K}$ such that:*

1. *$d = O(\min\{s \log(K|\mathcal{S}|)), s^2 \log(K)\})$*
2. *For every label $y$, we have $\|\mathbf{v}_y\|_2 = 1$.*
3. *For all $\mathbf{x} \in \mathcal{S}$ and $y \in L_{\mathbf{y}(\mathbf{x})}$, we have $\phi(\mathbf{x})^\top \mathbf{v}_y \geq \frac{2}{3}$.*
4. *For all $\mathbf{x} \in \mathcal{S}$ and $y \notin L_{\mathbf{y}(\mathbf{x})}$, we have $\phi(\mathbf{x})^\top \mathbf{v}_y \leq \frac{1}{3}$.*
5. *For every pair of labels $y, y'$ with $y \neq y'$, we have $\mathbf{v}_y^\top \mathbf{v}_{y'} \leq \sqrt{\frac{2 \log(4K^2)}{d}}$.*
6. *For any $\mathbf{x} \in \mathcal{S}$, we have $\|\phi(\mathbf{x})\|_2 = O(s(\frac{\log(K)}{d})^{\frac{1}{4}})$.*

This theorem shows that in the limit of infinite model capacity for constructing the embedding map, there exists a low-dimensional embedding-based linear classifier that thresholds at $\frac{1}{2}$ and has perfect training accuracy. Furthermore, the label embeddings $\mathbf{v}_y$ are normalized to unit length. Since deep neural networks have been demonstrated to have excellent function approximation capabilities, this result naturally motivates a model architecture which uses a deep neural network to mimic the optimal infinitely expressive embedding map $\phi$, followed by a linear classifier. Another consequence of the bound on the dimension in terms of $|\mathcal{S}|$ is it shows how overfitting is possible with small training sets: the dependence of the dimension $d$ on $s$ improves to linear from quadratic at the price of a (mild) logarithmic factor in the size of the sample set. On the other hand, applying the theorem with $\mathcal{S} = \mathcal{X}$ shows that $d = O(s^2 \log(K))$ suffices to obtain a classifier with perfect *test accuracy*.

## 3 Regularizing Embedding-Based Models

Motivated by our findings, in this section we propose a novel regularization framework and discuss its effectiveness for the classification problem with large output spaces.

### 3.1 Embedding Normalization

We first apply weight normalization proposed in [26]. In each layer, weight vectors of all output neurons share a single trainable length and each weight vector maintains its own trainable direction. Weight normalization not only helps stabilize training and accelerate convergence, but also improves generalization. For the ease of exposition, we assume all label embeddings are $\ell_2$-normalized to unit norm, i.e., $\mathbf{v}_i \in \mathbb{S}^{d-1}$, where $\mathbb{S}^{d-1}$ denotes the unit sphere in $\mathbb{R}^d$. In a similar vein, we can assume all input embeddings are normalized as well: $\phi_{\mathbf{w}}(\mathbf{x}) \in \mathbb{S}^{d-1}$. Our regularizer can be easily generalized to cases where the label embeddings are not unit norm.

### 3.2 GLaS Regularizer

In large-scale multi-label classification, the output space is both large and sparse — most feature vectors are associated with only very few true labels. Thus it may be desirable for an embedding-based classifier to have near-orthogonal label embeddings, as suggested by Theorem 2.1. As a result, it is natural to consider regularizers such as spread-out [37] that explicitly promote such structure.

**Spread-out Regularization.** Zhang et al. [37] introduced the spread-out regularization technique, which encourages local feature descriptors of images to be uniformly dispersed over the sphere. We consider a variant of spread-out regularization that brings the inner product of the embeddings of two different labels close to zero, i.e., $\mathbf{v}_y^\top \mathbf{v}_{y'} \approx 0$ if $y \neq y'$. More formally, the spread-out regularizer corresponds to the following:

$$\ell_{\text{spreadout}} = \frac{1}{K^2} \sum_{y=1}^{K} \sum_{y'=1}^{K} (\mathbf{v}_y^\top \mathbf{v}_{y'})^2. \tag{2}$$

Note that due to embedding normalization, diagonal entries $\mathbf{v}_y^\top \mathbf{v}_y = 1$ and hence these terms will not play a role in the regularization loss function in (2). Zhang et al. [37] have shown the effectiveness of this technique in learning good local feature descriptors for images. However, one major drawback of this regularizer is that it over-penalizes the embeddings of two different labels that occur frequently together (e.g., *apple* and *fruit* tend to co-occur for many inputs). In other words, label embeddings of labels that co-occur frequently are also encouraged to be far away, which is clearly undesirable.

**Correcting Over-penalization: GLaS Regularization.** The spread-out regularizer suffers from the lack of modeling the co-occurrences of labels. Thus, to correct for this over-penalization, we need to estimate the degree of occurrence between labels from training data and explicitly model it with the regularizer.

Let $Y \in \{0, 1\}^{n \times K}$ be the training set label matrix where each row corresponds to a single training example. Let $A = Y^\top Y$ so that $A_{y,y'}$ = number of times labels $y$ and $y'$ co-occur, and let $Z = \mathrm{diag}(A) \in \mathbb{R}^{K \times K}$ be the matrix containing only the diagonal component of $A$. Observe that $AZ^{-1}$ represents the conditional frequency of observing one label given the other. Indeed,

$$(AZ^{-1})_{y,y'} = \frac{A_{y,y'}}{A_{y',y'}} = \frac{\text{number of times } y \text{ and } y' \text{ co-occur}}{\text{number of times } y' \text{ occurs}} =: F(y|y').$$

Similarly, $Z^{-1}A = (AZ^{-1})^\top$ contains the conditional frequencies in reverse, i.e., $(Z^{-1}A)_{y,y'} = F(y'|y)$. These conditional frequencies encode the degree of co-occurrence between labels $y$ and $y'$, and we would like their embeddings $\mathbf{v}_y$ and $\mathbf{v}_{y'}$ to reflect this co-occurrence pattern:

$$\ell_{\mathrm{GLaS}} = \frac{1}{K^2} \left\| \mathbf{V}^\top \mathbf{V} - \frac{1}{2}(AZ^{-1} + Z^{-1}A) \right\|_F^2. \tag{3}$$

In the case where all labels are uncorrelated, this loss recovers the spread-out regularizer. While we choose to define the degree of label correlation as the average of conditional frequencies between labels, other measures of similarity such as pointwise mutual information (PMI) and Jaccard distance can also be used. In Appendix B, we give a theoretical justification for using the *geometric mean* of the conditional frequencies (see Theorem B.1). In experiments, however, we found empirically that using *arithmetic mean* of the conditional frequencies gives a slight but noticeable boost in accuracy compared to other measures, motivating the definition (3) of the GLaS regularizer.

One issue that arises when using this regularizer is that calculating $\ell_{\mathrm{GLaS}}$ requires $O(K^2)$ operations and becomes prohibitively expensive when $K$ is large. Instead, we select a batch of rows from $\mathbf{V}$ and compute a stochastic version of the loss on that batch only.

**Relationship to Graph Laplacian and Spread-out Regularization.** While the definition for the GLaS regularizer is intuitive, it may seem arbitrary and one can arrive at other regularizers by following a similar intuition. However, we show that the GLaS regularizer can be recovered as a sum of the well-known graph Laplacian regularizer and the spread-out regularizer, thus giving our regularizer its name (**G**raph **L**aplacian **a**nd **S**preadout).

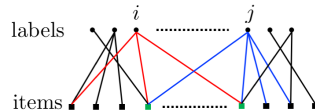

Figure 2: The item-label bipartite graph. The edge between a label node and an item node represents an assignment of the label to the item. Labels $i$ and $j$ have co-occurred in two items.

Graph Laplacian as a general technique has been successfully applied to representation learning problems such as metric learning [5] and hashing [21]. By adding a graph Laplacian based loss, we can impose the right structure on the off-diagonal values in the Gram matrix of label embeddings. More specifically, to assign similar embeddings to labels that co-occur frequently, we can explicitly penalize the $\ell_2$ distance between two label embeddings with a weight proportional to their co-occurrence statistics. As a result, the graph Laplacian regularization makes the label embeddings consistent with the connectivity pattern of label nodes in the item-label graph (Figure 2). We can write the graph Laplacian regularizer as

$$\ell_{\mathrm{Laplacian}} = \frac{1}{K^2} \sum_{y=1}^{K} \sum_{y'=1}^{K} \|\mathbf{v}_y - \mathbf{v}_{y'}\|_2^2 u_{yy'}, \tag{4}$$

where $u_{yy'}$ denotes the amount of "adjacency" between graph nodes of labels $y$ and $y'$ and is only dependent on the graph structure. However, this loss formulation admits a trivial optimal solution that assigns all labels the same embedding.

Recall that the spread-out regularizer suffers from a completely opposite weakness of encouraging all label embeddings to be orthogonal regardless of any correlation. Thus, combining the two regularizers has the effect of compensating their respective weaknesses and promoting their strengths. Summing the graph Laplacian regularizer (4) and the spread-out regularizer (2) we get

$$\ell_{\text{Laplacian}} + \ell_{\text{spreadout}} = \frac{1}{K^2} \sum_{y=1}^{K} \sum_{y'=1}^{K} \left[ \|\mathbf{v}_y - \mathbf{v}_{y'}\|_2^2 u_{yy'} + (\mathbf{v}_y^\top \mathbf{v}_{y'})^2 \right]$$

$$\stackrel{(a)}{=} \frac{1}{K^2} \sum_{y=1}^{K} \sum_{y'=1}^{K} \left[ (\mathbf{v}_y^\top \mathbf{v}_{y'} - u_{yy'})^2 - (u_{yy'}^2 - 2u_{yy'}) \right]$$

where (a) holds since $\|\mathbf{v}_y\|_2^2 = 1$. One can see that $\sum_{y,y'}(u_{yy'}^2 - 2u_{yy'})$ is a constant that only depends on the graph structure. The non-constant part of the sum can be written as $\frac{1}{K^2}\|\mathbf{V}^\top \mathbf{V} - U\|_F^2$, which is exactly the form of GLaS given in (3) with $U = \frac{1}{2}(AZ^{-1} + Z^{-1}A)$ being the measure of degree of adjacency in the label graph. Note that the graph Laplacian regularizer $\ell_{\text{Laplacian}}$ encourages frequently co-occurring labels to have similar label embeddings. However, labels that do not co-occur frequently but have similar embeddings are not penalized by graph Laplacian regularizer. This is achieved through the spread-out regularizer $\ell_{\text{spreadout}}$. Thus, our regularizer GLaS captures the essence of label relation.

---

**Algorithm 2** Training with regularization

---

1: **Input:** Dataset $\{(\mathbf{x}_1, \mathbf{y}_1), \dots, (\mathbf{x}_n, \mathbf{y}_n)\}$
2:　　　Feature embedding model $\phi_\mathbf{w} : \mathcal{X} \to \mathbb{R}^d$
3:　　　Label embedding matrix $\mathbf{V} \in \mathbb{R}^{d \times K}$
4:　　　Loss function $\ell : \mathbb{R}^K \times \mathcal{Y} \to \mathbb{R}$
5:　　　GLaS loss $\ell_{\text{GLaS}} : \mathbb{R}^{B \times B} \times \mathbb{R}^{B \times B} \to \mathbb{R}$
6:　　　Regularization weight $\lambda$
7:　　　Dropout probability $\rho \in [0, 1]$
8:　　　Learning rates $\eta_\mathbf{w}, \eta_\mathbf{V}$
9: Initialize $\mathbf{w}, \mathbf{V}$
10: **repeat**
11:　　Sample a batch $\mathbf{x}_1, \dots, \mathbf{x}_B$
12:　　Sample labels $y_1, \dots, y_B$ uniformly from non-zero indices of $\mathbf{y}_1, \dots, \mathbf{y}_B$
13:　　Apply input dropout $\mathbf{x}_i \leftarrow \mathbf{x}_i \odot \text{Bernoulli}(\rho, D)$
14:　　Compute loss $L \leftarrow \frac{1}{B} \sum_{i=1}^{B} \ell(\phi_\mathbf{w}(\mathbf{x}_i)^\top \mathbf{V}, y_i)$
15:　　$Y \leftarrow [\mathbf{y}_1 | \cdots | \mathbf{y}_B]$
16:　　$U \leftarrow B \times B$ submatrix of Equation (3) corresponding to indices $y_1, \dots, y_B$
17:　　$\mathbf{V} \leftarrow [\mathbf{v}_{y_1} | \cdots | \mathbf{v}_{y_B}] \in \mathbb{R}^{B \times B}$
18:　　Regularize $L \leftarrow L + \lambda \ell_{\text{GLaS}}(\mathbf{V}^\top \mathbf{V}, U)$
19:　　Compute gradients $\frac{dL}{d\mathbf{w}}$ and $\frac{dL}{d\mathbf{V}}$ via backpropagation
20:　　Update $\mathbf{w} \leftarrow \mathbf{w} - \eta_\mathbf{w} \frac{dL}{d\mathbf{w}}, \mathbf{V} \leftarrow \mathbf{V} - \eta_\mathbf{V} \frac{dL}{d\mathbf{V}}$
21: **until** convergence

---

### 3.3 Input Dropout

Input dropout [13] is a simple regularization and data augmentation technique for text classification models with sparse features. For a selected keep probability $\rho \in [0, 1]$ and an input feature $\mathbf{x}$, the method produces an augmented input $\mathbf{x}' = \mathbf{x} \odot \text{Bernoulli}(\rho, D)$, where $\odot$ denotes element-wise multiplication. Thus, non-zero feature coordinates are set to zero with probability $1 - \rho$. This can be interpreted as data augmentation, where features in the input are uniformly removed with probability $1 - \rho$. It discourages the model from fitting spurious patterns in input features when training data is scarce and it also promotes the model to be robust to corruption of the input features. The complete learning algorithm that integrates all techniques described in this section is presented as Algorithm 2.

## 4 Experiments

In this section, we present experimental results of our method on several widely used extreme multi-label classification datasets: AMAZONCAT-13K, AMAZON-670K, WIKILSHTC-325K,

DELICIOUS-200K, EURLEX-4K, and WIKIPEDIA-500K. The statistics of these datasets is presented in Table 5 in the supplementary material.

**Ablation Study.** We begin by studying the performance of Algorithm 2 under different settings of its hyperparameters. In particular, we investigate variations in the regularization type and weight, input dropout, batch size, and embedding type and size. Table 1 shows the effects of different parameters on the performance of our method on the AMAZONCAT-13K dataset. We first list our base setting that we have derived through cross validation. In the base setting, we use GLaS regularizer (discussed in Sec. 3.2) with regularization weight $\lambda = 10$, input dropout with $\rho = 0.8$, batch size $B = 4096$, and a non-linear embedding map $\phi_{\mathbf{w}}$ with embedding dimension $d = 1024$. In each row of Table 1, we alter one parameter from the base setting to study its impact. For the regularization method, we compare our method with the

| Variable | Parameters | | | |
|---|---|---|---|---|
| Regularizer | None | GLaS | Spread-out | Gravity |
| | 92.34 | **94.21** | 93.34 | 93.42 |
| Regularization Weight | $\lambda = 1$ | $\lambda = 10$ | | $\lambda = 100$ |
| | 93.68 | **94.21** | | 93.75 |
| Input Dropout | $\rho = 1.0$ | $\rho = 0.8$ | | $\rho = 0.6$ |
| | 93.39 | **94.21** | | 94.08 |
| Batch Size | 1024 | 2048 | | 4096 |
| | 94.04 | 93.98 | | **94.21** |
| Embedding Size | $d = 256$ | $d = 512$ | | $d = 1024$ |
| | 93.24 | 93.82 | | **94.21** |
| Embedding Type | Linear | Nonlinear (ReLU) | | |
| | 91.77 | **94.21** | | |

Table 1: Sensitivity of Algorithm 2 to variations in different parameters for AMAZONCAT-13K. Each row shows the effect of a single parameter. Our GLaS regularizer outperforms spread-out and gravity. A moderate regularization weight and input dropout, a large embedding size, and using nonlinearity lead to a better result.

spread-out regularizer [37] and Gravity regularizer [16] and show that our method significantly outperforms these two. We can observe that the regularization weight and input dropout rate should not be either excessively small or large as these settings hurt the test accuracy.

As one can expect, embeddings of higher dimensionalities outperform those of lower dimensionalities. Batch sizes in the range of 1000s do not have a significant impact on the performance; however, we do note that the largest batch size 4096 gives us the highest test accuracy. Finally and as shown in Table 1, adding the ReLU nonlinearity boosts the performance of $\phi_{\mathbf{w}}$ in learning the embedding.

**Generalization Gap.** As discussed previously, one of the main goals of this paper is to propose regularization techniques that mitigate the overfitting (Figure 1) of neural network embedding-based methods for extreme multi-label classification problems. Table 2 studies the effect of our regularization technique on the generalization gap, i.e., the difference between training and test accuracies. In particular, we have studied two datasets AMAZONCAT-13K and AMAZON-670K in two different settings:

| Dataset | Regularization | Train Acc. | Test Acc. | Gen. Gap |
|---|---|---|---|---|
| AMAZONCAT-13K | GLaS | 98.77 | 94.21 | **4.56** |
| | None | 99.23 | 92.34 | 6.89 |
| AMAZON-670K | GLaS | 96.10 | 46.32 | **49.78** |
| | None | 98.21 | 44.53 | 53.68 |

Table 2: The comparison of generalization gap in Algorithm 1 and Algorithm 2 when they are applied to AMAZONCAT-13K and AMAZON-670K datasets. The GLaS regularizer (Section 3.2) significantly improves the generalization gap.

with and without the regularization technique we discussed in Section 3. The table shows that regularizing embedding based models with our method reduces the generalization gap over the unregularized setting while improving test accuracy. As an example, GLaS regularizer reduces the generalization gap of Algorithm 1 by more than 30% when applied to the AMAZONCAT-13K dataset.

**Comparison with Previous Work.** We compare our method with several other recent works on the extreme classification problem denoted in Table 3. As shown in this Table, on all datasets except Delicious-200K and EURLex-4K our method matches or outperforms all previous work in terms of precision@k[3]. Even on the Delicious-200K dataset, our method's performance is close to that of the state-of-the-art, which belongs to another embedding-based method SLEEC [6]. One thing to note about the Delicious-200k dataset is that its average number of labels per training point is significantly larger than that of other datasets. Due to this, we observed that it took a long time for training to show steady progress with the fixed margin loss. Hence, we have used the softmax-cross-entropy loss for the Delicious-200K dataset instead of the loss function in (1). Softmax-cross-entropy loss relaxes the margin requirement and significantly stabilizes training.

|  |  | Embedding-Based | | | | | Other Methods | | | | | |
|---|---|---|---|---|---|---|---|---|---|---|---|---|
| Dataset | P@k | Ours | SLEEC [6] | LEML [36] | RobustXML [31] | XML-CNN [19] | PfastreXML [14] | FastXML [24] | Parabel [23] | DiSMEC [3] | PD-Sparse [34] | PPD-Sparse [33] |
| AMAZONCAT-13K | P@1 | *94.21* | 90.53 | - | 88.4 | - | 91.75 | 93.11 | 93.03 | 93.40 | 90.60 | - |
|  | P@3 | **79.70** | 76.33 | - | 74.6 | - | 77.97 | 78.2 | 79.16 | 79.10 | 75.14 | - |
|  | P@5 | **64.84** | 61.52 | - | 60.6 | - | 63.68 | 63.41 | 64.52 | 64.10 | 60.69 | - |
| WIKILSHTC-325K | P@1 | *65.46* | 54.83 | 19.82 | 53.5 | - | 56.05 | 49.75 | 65.04 | 64.40 | 61.26 | 64.08 |
|  | P@3 | *45.44* | 33.42 | 11.43 | 31.8 | - | 36.79 | 33.10 | 43.23 | 42.50 | 39.48 | 41.26 |
|  | P@5 | *34.51* | 23.85 | 8.39 | 29.9 | - | 27.09 | 24.45 | 32.05 | 31.50 | 28.79 | 30.12 |
| AMAZON-670K | P@1 | *46.38* | 35.05 | 8.13 | 31.0 | 35.39 | 39.46 | 36.99 | 44.89 | 44.70 | - | 45.32 |
|  | P@3 | *42.09* | 31.25 | 6.83 | 28.0 | 31.93 | 35.81 | 33.28 | 39.80 | 39.70 | - | 40.37 |
|  | P@5 | *38.56* | 28.56 | 6.03 | 24.0 | 29.32 | 33.05 | 30.53 | 36.00 | 36.10 | - | 36.92 |
| DELICIOUS-200K | P@1 | 46.4 | *47.85* | 40.73 | 45.0 | - | 41.72 | 43.07 | 46.97 | 45.50 | 34.37 | - |
|  | P@3 | 40.49 | *42.21* | 37.71 | 40.0 | - | 37.83 | 38.66 | 40.08 | 38.70 | 29.48 | - |
|  | P@5 | 38.1 | *39.43* | 35.84 | 38.0 | - | 35.58 | 36.19 | 36.63 | 35.50 | 27.04 | - |
| EURLEX-4K | P@1 | 77.5 | **79.26** | 63.4 | - | 76.38 | 75.45 | 71.36 | 81.73 | 82.4 | 76.43 | *83.83* |
|  | P@3 | 65.01 | 64.3 | 50.35 | - | 62.81 | 62.7 | 59.9 | 68.78 | 68.5 | 60.37 | *70.72* |
|  | P@5 | 54.37 | 52.33 | 41.28 | - | 51.41 | 52.51 | 50.39 | 57.44 | 57.7 | 49.72 | *59.21* |
| WIKIPEDIA-500K | P@1 | 69.91 | 48.2 | 41.3 | - | 59.85 | 59.52 | 54.1 | 66.73 | *70.2* | - | 70.16 |
|  | P@3 | **49.08** | 29.4 | 30.1 | - | 39.28 | 40.24 | 35.5 | 47.48 | *50.6* | - | 50.57 |
|  | P@5 | **38.35** | 21.2 | 19.8 | - | 29.81 | 30.72 | 26.2 | 36.78 | *39.7* | - | 39.66 |

Table 3: Performance comparison (based on precision@k) with several other methods on large-scale datasets. Our method attains or improves upon the state-of-the-art results. Results of other methods are derived from the extreme classification repository. Italic underlined numbers are the best of the entire row and bold numbers are the best among embedding-based methods.

|  |  | Embedding-Based | | | Other Methods | | | | | |
|---|---|---|---|---|---|---|---|---|---|---|
| Dataset | PSP@k | Ours | SLEEC [6] | LEML [36] | PfastreXML [14] | FastXML [24] | Parabel [23] | DiSMEC [3] | PD-Sparse [34] | PPD-Sparse [33] |
| AMAZONCAT-13K | PSP@1 | **47.53** | 46.75 | - | *69.52* | 48.31 | 50.93 | 59.10 | 49.58 | - |
|  | PSP@3 | 62.74 | 58.46 | - | *73.22* | 60.26 | 64.00 | 67.10 | 61.63 | - |
|  | PSP@5 | 71.66 | 65.96 | - | *75.48* | 69.30 | 72.08 | 71.20 | 68.23 | - |
| WIKILSHTC-325K | PSP@1 | *46.22* | 20.27 | 3.48 | 30.66 | 16.35 | 26.76 | 29.1 | 28.34 | 27.47 |
|  | PSP@3 | *46.15* | 23.18 | 3.79 | 31.55 | 20.99 | 33.27 | 35.6 | 33.50 | 33.00 |
|  | PSP@5 | *47.28* | 25.08 | 4.27 | 33.12 | 23.56 | 37.36 | 39.5 | 36.62 | 36.29 |
| AMAZON-670K | PSP@1 | *38.94* | 20.62 | 2.07 | 29.30 | 19.37 | 25.43 | 27.8 | - | 26.64 |
|  | PSP@3 | *39.72* | 23.32 | 2.26 | 30.80 | 23.26 | 29.43 | 30.6 | - | 30.65 |
|  | PSP@5 | *41.24* | 25.98 | 2.47 | 32.43 | 26.85 | 32.85 | 34.2 | - | 34.65 |
| DELICIOUS-200K | PSP@1 | *28.68* | 7.17 | 6.06 | 3.15 | 6.48 | 7.25 | 6.5 | 5.29 | - |
|  | PSP@3 | *24.93* | 8.16 | 7.24 | 3.87 | 7.52 | 7.94 | 7.6 | 5.80 | - |
|  | PSP@5 | *23.87* | 8.96 | 8.10 | 4.43 | 8.31 | 8.52 | 8.4 | 6.24 | - |
| EURLEX-4K | PSP@1 | *49.77* | 34.25 | 24.10 | 43.86 | 26.62 | 36.36 | 41.20 | 36.28 | - |
|  | PSP@3 | *51.05* | 38.35 | 26.37 | 45.23 | 32.07 | 41.95 | 44.30 | 40.96 | - |
|  | PSP@5 | *53.82* | 40.30 | 27.62 | 46.03 | 35.23 | 44.78 | 46.90 | 42.84 | - |

Table 4: Performance comparison (based on propensity scored precision@k, PSP@k) with several other methods on large-scale datasets. Propensity weights are higher for rarer labels, hence this metric better reflects the model's ability to generalize to tail labels than precision. Italic underlined numbers are the best of the entire row and bold numbers are the best among embedding-based methods.

One of the biggest challenges for learning in large output spaces comes from tail labels that are only assigned to a few inputs, but make up the majority of the whole label set. The propensity scored precision@K (PSP@K[4]) metric corrects for this bias by up-weighting rare labels. To demonstrate the effectiveness of our method at predicting tail labels, we report results using this evaluation metric in Table 4. While many previous methods that we compare against have to explicitly change their training objective or algorithm accordingly to account for the re-weighting, in contrast, our simple embedding based models learn to predict these tail labels remarkably well *without any adjustment* of our training loss or procedure. On the dataset with the largest number of labels Amazon-670K, our method improves the PSP@1 metric by an absolute margin of 9.6%.

**Training and Inference Speed.** We train all models up to 10 epochs and apply early stopping when evaluation accuracy ceases to improve. Though the overall training process takes minutes to hours, the time complexity is $O(d \sum_{\mathbf{x} \in \mathcal{S}} \text{nnz}(\mathbf{x}))$, where $d$ is the embedding dimensionality, $\mathcal{S}$ is the set of training samples, and $\text{nnz}(\mathbf{x})$ is the number of non-zero features of the sparse input $\mathbf{x}$.

At inference time, we apply efficient Maximum Inner Product Search techniques such as [11, 30]. The non-exhaustive search achieves low latency due to highly effective clustering based tree indices [2] and hardware based acceleration [11, 8]. For all datasets up to a few million labels, the inference latency is below 10ms and below 1ms for under 100k labels.

## 5 Conclusions

In this paper, we showed that from both theoretical and empirical perspectives, neural network models suffer from *overfitting* instead of low-dimensional embedding bottleneck when applied to extreme multi-label classification problems. To this end, we introduced the GLaS regularization framework and demonstrated its effectiveness with new state-of-the-art results on several widely tested large-scale datasets. We hope future work can build on our theoretical and empirical findings and more competitive embedding-based methods can be developed along this direction.

## Footnotes

[3]$P@k = \frac{1}{k} \sum_{l \in \mathrm{rank}_k(\hat{\mathbf{y}})} \mathbf{y}_l$ where $\hat{\mathbf{y}}$ is the predicted score vector and $\mathbf{y} \in \{0,1\}^L$ is the ground truth labels.

[4]Similar to P@k, PSP@k $= \frac{1}{k} \sum_{l \in \text{rank}_k(\hat{\mathbf{y}})} \frac{\mathbf{y}_l}{p_l}$ where $p_l$ denotes the propensity weights.

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
