[Supplementary Material]

# A    Existence of Perfect Accuracy Low-Dimensional Embedding Classifiers

**Theorem A.1.** *Let $\mathcal{S} \subseteq \mathcal{X}$ be a sample set. Under the assumption on the data specified above, there exists a function $\phi : \mathcal{X} \to \mathbb{R}^d$, and a label embedding matrix $\mathbf{V} \in \mathbb{R}^{d \times K}$ such that:*

1. *$d = O(\min\{s \log(K|\mathcal{S}|)), s^2 \log(K)\})$*
2. *For every label $y$, we have $\|\mathbf{v}_y\|_2 = 1$.*
3. *For all $\mathbf{x} \in \mathcal{S}$ and $y \in L_{\mathbf{y}(\mathbf{x})}$, we have $\phi(\mathbf{x})^\top \mathbf{v}_y \geq \frac{2}{3}$.*
4. *For all $\mathbf{x} \in \mathcal{S}$ and $y \notin L_{\mathbf{y}(\mathbf{x})}$, we have $\phi(\mathbf{x})^\top \mathbf{v}_y \leq \frac{1}{3}$.*
5. *For every pair of labels $y, y'$ with $y \neq y'$, we have $\mathbf{v}_y^\top \mathbf{v}_{y'} \leq \sqrt{\frac{2 \log(4K^2)}{d}}$.*
6. *For any $\mathbf{x} \in \mathcal{S}$, we have $\|\phi(\mathbf{x})\|_2 = O(s(\frac{\log(K)}{d})^{\frac{1}{4}})$.*

*Proof.* We show the existence of the function $\phi$ and $\mathbf{V}$ using the probabilistic method. First, let $\mathbf{V}$ be chosen by sampling each entry uniformly at random in $\{-\frac{1}{\sqrt{d}}, \frac{1}{\sqrt{d}}\}$, where the exact specification of $d$ will be revealed in the subsequent analysis. Clearly, for all labels $y$, we have $\|\mathbf{v}_y\|_2 = 1$, which establishes item 2. For any $\mathbf{x} \in \mathcal{X}$, define $\phi(\mathbf{x}) = \sum_{y \in L_{\mathbf{y}(\mathbf{x})}} \mathbf{v}_y$. For any $\mathbf{x} \in \mathcal{X}$ and any label $y$, we have

$$\phi(\mathbf{x})^\top \mathbf{v}_y = \mathbb{I}[y \in L_{\mathbf{y}(\mathbf{x})}] + \sum_{y' \in L_{\mathbf{y}(\mathbf{x})}, y \neq y'} \mathbf{v}_y^\top \mathbf{v}_{y'}.$$

By an application of Hoeffding's inequality,

$$\Pr\left[\left|\sum_{y' \in L_{\mathbf{y}(\mathbf{x})}, y \neq y'} \mathbf{v}_y^\top \mathbf{v}_{y'}\right| > \tfrac{1}{3}\right] \leq 2 \exp(-\tfrac{d}{18s}).$$

Now note that $|\{\phi(\mathbf{x}) : \mathbf{x} \in \mathcal{S}\}| \leq \min\{|\mathcal{S}|, K^s\}$. Thus, by a union bound, we conclude that

$$\Pr\left[\exists \mathbf{x} \in \mathcal{S}, y : \left|\sum_{y' \in L_{\mathbf{y}(\mathbf{x})}, y \neq y'} \mathbf{v}_y^\top \mathbf{v}_{y'}\right| > \tfrac{1}{3}\right]$$
$$\leq 2 \min\{|\mathcal{S}|, K^s\} \exp(-\tfrac{d}{18s}).$$

By similar calculations, we also have, for any given $t > 0$,

$$\Pr[\exists y \neq y' : |\mathbf{v}_y^\top \mathbf{v}_{y'}| > t] \leq 2K^2 \exp(-\tfrac{dt^2}{2}).$$

Set $d = \lceil 18s \log(4 \min\{|\mathcal{S}|, K^s\}) \rceil$ (which establishes item 1) and $t = \sqrt{\frac{2 \log(4K^2)}{d}}$ so that the above two probabilities add up to less than 1. Thus, there exists a matrix $\mathbf{V}$ s.t. for all $\mathbf{x} \in \mathcal{S}$ and all $y$, $|\phi(\mathbf{x})^\top \mathbf{v}_y - \mathbb{I}[y \in L_{\mathbf{y}(\mathbf{x})}]| \leq \frac{1}{3}$ (which implies items 3 and 4) and for all pairs of labels $y \neq y'$, we have $|\mathbf{v}_y^\top \mathbf{v}_{y'}| \leq \sqrt{\frac{2 \log(4K^2)}{d}}$ (which implies items 5). Finally, for item 6, note that for any $\mathbf{x}$, we have

$$\phi(\mathbf{x})^\top \phi(\mathbf{x}) = \sum_{y \in L_{\mathbf{y}(\mathbf{x})}} \mathbf{v}_y^\top \mathbf{v}_y + \sum_{y, y' \in L_{\mathbf{y}(\mathbf{x})}, y \neq y'} \mathbf{v}_y^\top \mathbf{v}_{y'} \leq s + s(s-1) \cdot \sqrt{\frac{2 \log(4K^2)}{d}}. \qquad \square$$

# B    Theoretical Justification of the GLaS Regularizer

In this section we give theoretical justification for the definition of the GLaS regularizer (3). Specifically, we prove a representability theorem analogous to Theorem 2.1. This theorem shows that it is possible to construct a low-dimensional embedding-based classifier which corrrectly labels all examples in the training set, and additionally, the inner-products of the embeddings of each pair of labels are close to the *geometric means* of their conditional frequencies. The definition of the GLaS regularizer (3) uses the *arithmetic mean* of the conditional frequencies instead of the geometric means due to superior experimental performance (although the geometric-means-based regularizer has very similar performance).

We first recall some notation from Section 3.2. Suppose $\mathcal{S} \subseteq \mathcal{X}$ be a sample set of size $n$. Let $Y \in \{0, 1\}^{n \times K}$ be the training set label matrix where each row corresponds to a single training example. Let $A = Y^\top Y$ so that $A_{y,y'} =$ number of times labels $y$ and $y'$ co-occur, and let $Z = \text{diag}(A) \in \mathbb{R}^{K \times K}$ be the matrix containing only the diagonal component of $A$. The matrix $AZ^{-1}$ gives conditional frequencies of observing one label given another: $(AZ^{-1})_{y,y'} = F(y|y')$.

| Dataset | Feature Dimensionality | Label Dimensionality | Number of Train Points | Number of Test Points | Avg. Points Per Label | Avg. Labels Per Point |
|---|---|---|---|---|---|---|
| AMAZONCAT-13K | 203,882 | 13,330 | 1,186,239 | 306,782 | 448.57 | 5.04 |
| AMAZON-670K | 135,909 | 670,091 | 490,449 | 153,025 | 3.99 | 5.45 |
| WIKILSHTC | 1,617,899 | 325,056 | 1,778,351 | 587,084 | 17.46 | 3.19 |
| DELICIOUS-200K | 782,585 | 205,443 | 196,606 | 100,095 | 72.29 | 75.54 |
| EURLEX-4K | 5,000 | 3,993 | 12,920 | 3,185 | 25.73 | 5.31 |
| WIKIPEDIA-500K | 2,381,304 | 501,070 | 1,813,391 | 783,743 | 24.75 | 4.77 |

Table 5: Summary of the dataset statistics discussed in the paper.

**Theorem B.1.** *Suppose that for the sample set $\mathcal{S} \subseteq \mathcal{X}$, we have $A_{y,y} \leq a$ for some constant $a \ll n$. Let $\epsilon := \frac{1}{2\sqrt{a}}$. Then there exists a function $\phi : \mathcal{X} \to \mathbb{R}^d$, and a label embedding matrix $\mathbf{V} \in \mathbb{R}^{d \times K}$ such that:*

1. *$d = O(a \log(Kn))$*
2. *For any $\mathbf{x} \in \mathcal{S}$, we have $\|\phi(\mathbf{x})\|_2 < 1 + \epsilon$.*
3. *For every label $y$, we have $\|\mathbf{v}_y\|_2 < 1 + \epsilon$.*
4. *For all $\mathbf{x} \in \mathcal{S}$ and $y \in L_{\mathbf{y}(\mathbf{x})}$, we have $\phi(\mathbf{x})^\top \mathbf{v}_y > \epsilon$.*
5. *For all $\mathbf{x} \in \mathcal{S}$ and $y \notin L_{\mathbf{y}(\mathbf{x})}$, we have $\phi(\mathbf{x})^\top \mathbf{v}_y < \epsilon$.*
6. *For every pair of labels $y, y'$ we have $\left| \mathbf{v}_y^\top \mathbf{v}_{y'} - \sqrt{F(y|y')F(y'|y)} \right| < \epsilon$.*

*Proof.* Consider the following construction. For every $\mathbf{x} \in \mathcal{S}$, associate a unique standard basis vector $\mathbf{e}_\mathbf{x} \in \mathbb{R}^n$. Then, for every label $y$, define $\mathbf{v}'_y = \frac{1}{\sqrt{A_{y,y}}} \sum_{\mathbf{x} \in \mathcal{S}: y \in \mathbf{y}(\mathbf{x})} \mathbf{e}_\mathbf{x}$. It is easy to check, by direct calculation, the following properties:

1. For all labels $y$, we have $\|\mathbf{v}'_y\|_2 = 1$.
2. For all $\mathbf{x} \in \mathcal{S}$ and labels $y$, we have

$$
\mathbf{e}_\mathbf{x}^\top \mathbf{v}'_y = \begin{cases} \frac{1}{\sqrt{A_{y,y}}} \geq \frac{1}{\sqrt{a}} & \text{if } y \in L_{\mathbf{y}(\mathbf{x})} \\ 0 & \text{otherwise.} \end{cases}
$$

3. For all pairs of labels $y, y'$, we have $\mathbf{v}'_y{}^\top \mathbf{v}'_{y'} = \frac{A_{y,y'}}{\sqrt{A_{y,y} A_{y',y'}}} = \sqrt{F(y|y')F(y|y')}$.

Now, consider the Johnson-Lindenstrauss (JL) transform $\psi : \mathbb{R}^n \to \mathbb{R}^d$ with $d = O(\frac{\log(Kn)}{\epsilon^2}) = O(a \log(Kn))$ applied to the vectors $\mathbf{e}_\mathbf{x}$ for $\mathbf{x} \in \mathcal{S}$ and $\mathbf{v}'_y$ for labels $y$. Since these vectors are all unit length, by choosing a large enough constant in $O(\cdot)$ notation for $d$, the JL transform preserves all pairwise inner products of the vectors up to an additive error less than $\epsilon$. We now define $\phi(\mathbf{x}) = \psi(\mathbf{e}_\mathbf{x})$ for all $\mathbf{x} \in \mathcal{S}$ and $\mathbf{v}_y = \psi(\mathbf{v}'_y)$ for all labels $y$. Now the claims of the theorem follow immediately from the fact that the properties 1, 2 and 3 above are all preserved up to an error less than $\epsilon$. ☐

This theorem implies that there exists an embedding-based classifier which has perfect accuracy on the training set $\mathcal{S}$ when a threshold of $\epsilon$ is used. Furthermore, the label and input embeddings are nearly unit length, and the inner products of the label embeddings for each pair of labels are close to the geometric means of the conditional frequencies of the pair.

## C Additional Experimental Results

This section includes the summary of the dataset statistics that we have used in our experiments (Table 5). In addition, we have included a variant of precision, namely nDCG@k [4] results of different methods over different datasets (Table 6).

| Dataset | nDCG@k | Embedding-Based | | | | | Other Methods | | | | | |
|---|---|---|---|---|---|---|---|---|---|---|---|---|
| | | Ours | SLEEC [6] | LEML [36] | RobustXML [31] | XML-CNN [19] | PfastreXML [14] | FastXML [24] | Parabel [23] | DiSMEC [3] | PD-Sparse [34] | PPD-Sparse [33] |
| AMAZONCAT-13K | nDCG@1 | *94.21* | 90.53 | - | - | - | 91.75 | 93.11 | 93.03 | 93.40 | 90.60 | - |
| | nDCG@3 | *88.06* | 84.96 | - | - | - | 86.48 | 87.07 | 87.72 | 87.70 | 84.00 | - |
| | nDCG@5 | *86.08* | 82.77 | - | - | - | 84.96 | 85.16 | 86.00 | 85.80 | 82.05 | - |
| WIKILSHTC-325K | nDCG@1 | *65.53* | 54.83 | 19.82 | 53.5 | - | 56.05 | 49.75 | 65.04 | 64.40 | 61.26 | - |
| | nDCG@3 | **57.92** | 47.25 | 14.52 | 46.0 | - | 50.59 | 45.23 | *59.15* | 58.50 | 55.08 | - |
| | nDCG@5 | **57.09** | 46.16 | 13.73 | 44.0 | - | 50.13 | 44.75 | *58.93* | 58.40 | 54.67 | - |
| AMAZON-670K | nDCG@1 | *46.32* | 34.77 | 8.13 | 31.0 | 35.39 | 39.46 | 36.99 | 44.89 | 44.70 | - | - |
| | nDCG@3 | *44.36* | 32.74 | 7.30 | 28.0 | 33.74 | 37.78 | 35.11 | 42.14 | 42.10 | - | - |
| | nDCG@5 | *42.84* | 31.53 | 6.85 | 26.0 | 32.64 | 36.69 | 33.86 | 40.36 | 40.50 | - | - |
| DELICIOUS-200K | nDCG@1 | 46.4 | *47.85* | 40.73 | 45.0 | - | 41.72 | 43.07 | 46.97 | 45.50 | 34.37 | - |
| | nDCG@3 | 41.83 | *43.52* | 38.44 | 40.0 | - | 38.76 | 39.70 | 41.72 | 40.90 | 30.60 | - |
| | nDCG@5 | 39.7 | *41.37* | 37.01 | 37.0 | - | 37.08 | 37.83 | 39.07 | 37.80 | 28.65 | - |
| EURLEX-4K | nDCG@1 | 77.69 | **79.26** | 63.4 | - | 76.38 | 75.45 | 71.36 | 81.73 | *82.4* | 76.43 | - |
| | nDCG@3 | **68.16** | 68.13 | 53.56 | - | 66.28 | 65.97 | 62.87 | 72.15 | *72.50* | 64.31 | - |
| | nDCG@5 | **62.71** | 61.60 | 48.47 | - | 60.32 | 60.78 | 58.06 | 66.40 | *66.70* | 58.78 | - |
| WIKIPEDIA-500K | nDCG@1 | *69.91* | - | - | - | 59.85 | - | - | - | - | - | - |
| | nDCG@3 | *58.87* | - | - | - | 48.67 | - | - | - | - | - | - |
| | nDCG@5 | *56.32* | - | - | - | 46.12 | - | - | - | - | - | - |

Table 6: Performance comparison (based on normalized Discounted Cumulative Gain, i.e., nDCG@k — a variant of precision) with several other methods on large-scale datasets. Our method attains or improves upon the state-of-the-art results. Results of other methods are derived from the extreme classification repository. Italic underlined numbers are the best of the entire row and bold numbers are the best among embedding based methods.

# D  Python Code for PSP@K

This section includes the Tensorflow code of PSP@K computation. Our Tensorflow code is based on the MATLAB code available in the extreme classification repository[5].

```python
def precision_wt_k(scores, labels, wts, k):
  """
  Args:
    labels: Tensor of 0/1 labels with shape [batch_size, #classes].
    scores: Tensor of scores with shape [batch_size, #classes].
    wts: inverse propensity weights
    K: as in p@k

  Returns:
    psp_k: PSP@K
  """

  idx = tf.where(tf.not_equal(labels, 0))
  wts_labels = tf.sparse.to_dense(
      tf.SparseTensor(indices=idx,
                      values=tf.gather(wts,
                                       tf.cast(idx[:,1],tf.int64)),
                      dense_shape=labels.shape))
  psp_num = psp_precision(labels, scores, k, wts)
  psp_denum = psp_precision(labels, wts_labels, k, wts)
  psp_k = tf.divide(psp_num, psp_denum)

  return psp_k

def psp_precision(labels, scores, K, wts):
  """
  Args:
    labels: Tensor of 0/1 labels with shape [batch_size, #classes].
    scores: Tensor of scores with shape [batch_size, #classes].
    K: as in p@k
    wts: inverse propensity weights

  """

  _, indices = tf.math.top_k(tf.cast(scores, tf.float32), k=K)
  first_column = tf.reshape(
      tf.transpose(
          tf.tile(
              tf.expand_dims(
```

[5]http://manikvarma.org/downloads/XC/XMLRepository.html

```
41                       tf.range (0, tf.shape (indices) [0]) ,0) ,[K, 1])) ,[ -1])
42     sparse_indices = tf.stack ([ first_column ,
43                                 tf.reshape (indices , [ -1])], axis =1)
44
45     expanded_weights = tf.gather (wts ,
46                                   tf.cast (sparse_indices [:, 1] ,
47                                       tf.int64))
48     topK_mat = tf.SparseTensor (indices=tf.cast (sparse_indices ,
49                                           tf.int64) ,
50                                 values=tf.cast (expanded_weights ,
51                                           tf.float32) ,
52                                 dense_shape=tf.shape (labels ,
53                                           out_type=tf.int64))
54     topK_mat = tf.sparse.reorder (topK_mat)
55     prod = tf.multiply (tf.sparse.to_dense (topK_mat) ,
56                       tf.cast (labels , tf.float32))
57
58     return tf.reduce_mean (tf.divide (tf.reduce_sum (prod ,1) , K))
59
60
61  def psp_wts (labels , A=0.55 , B=1.5) :
62    """ Computes propensity weights for the NxK full test label matrix.
63        Wiki -LSHTC: A = 0.5, B = 0.4
64        Amazon: A = 0.6, B = 2.6
65        Others (default): A = 0.55 , B = 1.5
66
67    Args :
68      labels: is the binary matrix of (all) true labels
69      A: dataset -dependent constant
70      B: dataset -dependent constant
71
72    Returns :
73      wts: inverse propensity weights
74    """
75    N = labels.dense_shape [0]
76    counts = tf.cast (tf.sparse.reduce_sum (labels , 0) , tf.float32)
77    C = (tf.log (tf.cast (N,tf.float32)) - 1) * math.pow (B + 1, A)
78    wts = 1 + tf.multiply (C, tf.pow (counts+B,-A))
79    return wts
```

## Footnotes

[4] $\text{nDCG@k} = \frac{DCG@k}{\sum_{l=1}^{\min(k, \|\mathbf{y}\|_0)} \frac{1}{\log(l+1)}}$ where $\text{DCG@k} = \sum_{l \in \text{rank}_k(\hat{\mathbf{y}})} \frac{\mathbf{y}_l}{\log(l+1)}$, $\hat{\mathbf{y}}$ is the predicted score vector and $\mathbf{y} \in \{0, 1\}^L$ is the ground truth labels.