[Reviews · NeurIPS 2019]

Reviewer 1



In the prior literature, they cited the low dimensional embedding methods is the reason of the poor performance of the embedding based methods. In this paper, the author proposed that the final score vector for the labels actually generated by highly non-linear transformation such as thresholding the scores. Thus it is not clear if the low-rank structure of the score vectors directly cause the low-rank on the label vectors. Furthermore, the author uses a simple neural network to mimic the low-dimensional embedding can attain near-perfect training accuracy but generalize poorly and suggesting that overfitting is the root cause of the poor performance of the embedding based methods. This is the first contribution of the paper which breaks the glass ceiling of embedding based methods. In order to address the overfitting problem, it highlights the need of regularization techniques. We know that most feature vectors associated with only very few true labels, thus the label embedding matrix should have near-orthogonal structure, which is exactly what the Spread-out regularizer did. But the drawback is that it ignores the frequently co-occur labels correlation, thus it is not desirable. In order to address the lack of modeling co-occurrences of labels bring by spread-out regularizer, the author estimates the degree of occurrences between labels from the training data. The author built a co-occurrence label matrix and encode the conditional frequencies between labels into the loss function, which is the novel framework of GLaS Regularization. This method defines the degree of label correlation as the arithmetic mean of conditional frequencies between labels. In order to get rid of the expensive computation, the author chooses to compute stochastic loss on a specific batch. The GLaS regularizer’s name actually comes from the sum of graph Laplacian regularizer and the spread-out regularizer. Graph Laplacian regularizer tends to assign similar embeddings to labels that co-occur frequently, but the drawback is that it doesn’t penalize the similar embeddings of the non-related labels. This drawback can be solved by spread-out regularizer which encourages all label embeddings to be orthogonal regardless of any correlation. Thus, summing the graph Laplacian regularizer and the spread-out regularizer, the author proposes their regularizer which captures the advantages of both regularizer. Through the experiments on several widely used extreme (millions of labels) multi-label classification datasets, the proposed regularizer method matches or outperforms with all previous work in terms of precision on most dataset. The main contribution of this paper is to point out the bottleneck of the inferior performance of embedding-based method is not limited to the low-dimensional embedding problem, instead, the author proves the poor performance is because of the overfitting issue, and proposes their novel regularizer which is the sum of graph Laplacian regularizer and the spread-out regularizer and evaluate the effectiveness on several widely used datasets. From this point of view, research can focus on finding the other better regularizer and it is feasible to use low-dimensional embedding based methods even with simple linear classifiers to reach competitive result.

Reviewer 2



After the rebuttal: - Thanks for the plot. That is useful to see and in favor of the paper. - Note that representability based on universal approximation theorem does not imply learnability. Finding those representative weights, can still be NP hard. ------------- This paper addresses the problem of extreme multi-label classification. The paper proves that the embedding model is expressive. It then provides an example of experiment on data where the embedding-based model performed poorly as the result of overfitting. Once authors diagnose the problem as overfitting, they propose regularizers to constrain the model and improve the situation. Strengths: - The paper addresses an important and relevant research question: is extreme multi-label classification learnable by embedding models? - The expressiveness theory (Section 2.2) is useful. Weaknesses: - Quality: Results of Section 2.1, which builds the main motivation of the paper, is demonstrated on a very limited settings and examples. It does not convince the reader that overfitting is the general reason for potential poor performance of the models under study. - Soundness: While expressiveness is useful, it does not mean that the optimal weights are learnable. The paper seem to not pay attention to this issue. - Clarity: Related work could be improved. Some related works are mainly named but their differences are not described enough. - Organization could be improved. Currently the paper is dependent on appendix (eg the algorithms). Also the contents of tables are too small. Overall, I do not think the quality of the paper is high enough and I vote for it to be rejected.

Reviewer 3



==============Post rebuttal ========= Thanks to the authors for their feedback. Somethings still remain unclear : 1. Label embedding methods for large output spaces can be seen as deep networks with one hidden layer. It seems that the main message of the paper is that training these as deep nets with corresponding optimization schemes, rather than methods from conventional label embedding methods such as LEML and SLEEC leads to better results. Instead, the paper's main message conveys something different - namely, over-fitting of label embedding is the main cause for label-embedding methods, and it is fixed by regularization. Instead, on these datasets, a linear classifier also overfits, and it is not much of a surprise that a label embedding method/one hidden layer network overfits also. 2. It is somewhat difficult to understand the 200% increase in the propensity scored metrics(PSP@k in Table 4) over other sota baselines, while only below par or similar performance on vanilla P@k (Table 3). For instance, compare the proposed method vs SLEEC for various datasets. Does it mean that the method is working very poorly on head labels that it deriving all its performance from the tail-labels. In which case, it should be verified that of the model is indeed is doing poorly on head labels, and needs to be investigated why that is happening. 3. There should be much more clarity on the training time in terms of number of cores, and number of hours/days it took on diffrenent datasets, since this is a standard practice in XMC community, which makes it comparable to other methods. ================= The paper presents a simple deep learning method with one hidden layer for learning with large output spaces. The results show that the proposed method can achieve comparable performance to state-of-the-art linear methods such as DiSMEC/PPDSparse and tree-based methods such as Parabel. On a high level, I have the following concerns about the paper : 1. The paper suggests that they have found that overfitting is the main cause for bad performance of label-embedding methds, and that this is fixed by Glas regularizer. However, that does not seem to be the case, as from Table 1 and Table 2, the results without the Glas regularization are almost in the same ball-park as state-of-the-art methods, and the regularization merely helps by couple of % points. It seems that the main idea is that take a simple deep network (fully connected) with one hidden layer(acting as embedding) it gives close to sota performance, and one can increase the performance somewhat by regularization 2. It should be discussed why LEML (and SLEEC) which is based on similar scheme architecture but does not train it as a deep network works so poorly while the proposed method works better. Is it due to different optimization algorithm scheme. 3. Over-fitting is not new for these datasets, and it is likely that even linear classifiers such as DiSMEC overfits most of these datasets. For instance, one can get close to 99% training set accuracy with DiSMEC on Amazon13k data. So, it seems that showing over-fitting with embedding scheme is not very surprising. 4. The results for propensity score metrics (Table 4) seem rather surprising, since these metrics typically correlate very well with vanilla metrics (Table 3). Even though the vanilla metrics are very close to most sota methods, the prop scored metrics are almost double in some cases, which is really surprising. Since the proposed method does nothing special for tail-labels, and 70-90% are tail labels, the vanilla metric should also much higher, in that case. In this respect, it would be useful to be provided with the code to verify the claims, or the authors could check if their is correct, and if so, then investigate the reason for such large discrepancy. 5. The proof of Theorem 2.1 is not quite clear. In line 474 it seems that the proof if for some x that exists, and it becomes 477 it becomes for all x. The transformation is not immediate, and unclear. 6. The paper does not say anything about the computing infrastructure used to train the models for big datasets. It is important for fair comparison since most other works provide some information oin this context. 7. Missing references 1. The paper does not compare with AttentionXML [1], another deep learning approach which seems to be giving quite good results. 2. There is no comparison with ProXML [2] which is a sota method for propensity scored metrics, even though the code is available. Moreover, it also mentions ideas related to using graph laplacian for label co-occurrence. [1] AttentionXML: Extreme Multi-Label Text Classification with Multi-Label Attention Based Recurrent Neural Networks, https://arxiv.org/abs/1811.01727 [2] Data scarcity, robustness and extreme multi-label classification, https://link.springer.com/content/pdf/10.1007%2Fs10994-019-05791-5.pdf

[Author Response · NeurIPS 2019]

**Reviewer 1**: Thank you for the insightful analysis and acknowledgement of our effort.

**Reviewer 2**: **Re Quality**: Empirically we observe overfitting during training on all public datasets, which Reviewer 3 has also mentioned. We chose Amazon-13K as a representative dataset to demonstrate this phenomenon, but can certainly include plots on other datasets in the supplementary material for further evidence.

Figure 1: Training (blue) and test (red) accuracy on the EU-RLEX dataset when trained on the union of training and test data.

**Re Soundness**: Given enough capacity, a two-layer neural network (such as our model in Sec. 4) is theoretically guaranteed to be able to represent *any* continuous mapping since it is a universal approximator (Hornik, 1991). The question here is whether the embedding model will overfit to limited samples, which is our main message and is addressed by the GLaS regularizer.

**Re Clarity and organization:** Due to limited space, we could only include the most relevant prior works but can certainly include additional details in the supplementary. We moved less relevant details to the supplementary (including the pseudo-code) to include more experimental results. We will split the table to improve readability and reorganize the pseudo-code in the main text in the final version using the extra granted page if the paper gets accepted.

**Re Training on the union of training and test data**: Fig. 1 shows P@k across training epochs when we train on the union of training and test data. The model is clearly expressive enough as training and test accuracy are near-perfect.

**Reviewer 3**: **1.** XMC datasets have been well-researched and improvements ***"by couple of % points"*** are significant. For example, Parabel (WWW '18) only improved P@1 on WikiLSHTC(+0.64%) over the previous best but we significantly improved P@k SOTA on 3 datasets and PSP@k SOTA on 4 datasets. Regarding the use of deep learning, note that our method outperforms XML-CNN in Table 3 that uses both convolutional and fully-connected layers. Hence, our work is a matter of using an appropriate (even simple) architecture, loss function, optimization, and regularization. Our main goal is to debunk the low-dimensional bottleneck misconception by demonstrating this through a simple neural network model with a novel regularization framework.

**2.** It is true that LEML and SLEEC use similar architectures, but there are dramatic differences in the choice of the loss function: LEML uses a least square regression loss, whereas SLEEC uses a nearest neighbor loss. In contrast, our approach uses a margin-based loss complemented by stochastic optimization and novel regularization. While we agree that it is certainly informative to analyze where the crucial difference lies, we believe it is also important to highlight the main message of the paper, namely that proper design and training of embedding-based models can enable them to outperform other approaches.

**3.** We did not intend to make this impression that over-fitting is new or surprising for these datasets. Our goal was to show that we should not attribute the poor performance of embedding-based methods to the low-dimensional bottleneck, in direct response of the following quote from the DiSMEC paper, *"In XMC setting which consists of a diverse power-law distributed label space, the crucial assumption made by the embedding-based approaches of a low rank label space breaks down."* In Sec 2.2 and Theorem 2.1, we rigorously showed the existence of a perfect accuracy low-dimensional embedding-based classifier and the possibility of over-fitting with small training sets.

**4.** It is not necessary for PSP@k metrics to correlate very well with P@k and our results are not the first to be *"surprising."* **For example, compare P@k and PSP@k of PfastreXML and FastXML in Table 3 and Table 4. We also respectfully disagree with the reviewer's quote *"the proposed method does nothing special for tail-labels."*** In lines 204-206 we have mentioned that because of tail labels we regularize the label embeddings to be near-orthogonal by Eq. (2). Note that near-orthogonality is condition No. 5 mentioned in Theorem 2.1 for the existence of a perfect embedding-based classifier. GLaS regularization corrects over-penalizing based on the co-occurrence of labels which is indeed correlated with algebraic connectivity. As label co-occurrence or algebraic connectivity gets smaller (consider Amazon670k, WikiLSHTC, EURLex in Table 2 of arXiv:1803.01570), we get better PSP@k improvement over P@k because of having more near-orthogonal embedding and less GLaS correction. Due to the lack of space, we were not able to include our code here. However, our code is a TensorFlow translation of the MATLAB code provided in the XMC repository and we have verified its correctness and we will release it in the final version.

**5.** The values of $d$ and $t$ are chosen so that the sum of the probabilities in lines 474 and 475 is less than 1, which implies that with probability $> 0$, neither of the two events happens, from which the statement of line 478 follows.

**6.** We perform training on a cluster of servers with 2 Intel Xeon CPUs and inference with a single thread on a single server. We accelerate inference with approximate inner product search algorithms to bring the inference time to below 10 ms for large datasets. We are happy to provide more details if this is a point of concern.

**7.** We mainly relied on the XMC repository for the baselines but will for sure cite these references. Re ProXML, note that our PSP@k results outperform the ones in the ProXML paper. Re AttentionXML, it is a tree-based model that unlike all other baselines use raw text features and our method outperforms it on PSP@k metric (Fig 2 of AttentionXML paper on Amazon670k dataset). Also, please note that according to Hugo Larochelle (NeurIPS PC) *"it is not reasonable to compare current NIPS submissions with work that hasn't been accepted at a venue prior to submission."*

[Meta-Review · NeurIPS 2019]

There is some disagreement about the significance of the paper among the reviewers. Three steps can be distinguished. First, to refute the common belief that low-dimensional embeddings act as bottlenecks that limit the accuracy in the extreme classification case. Here, while it is true (raised by reviewer 1) that a representation result does not imply computational achievability, I feel that it reverses the direction of justification. If someone could show that common optimization methods fail to find embeddings (which "exist"), then this would re-instantiate the argument, yet in a more refined/precise form. I still feel that the paper makes an important argument in an ongoing discussion. Second, the finding that overfitting is the key problem. I agree here with reviewer 2 that overfitting has been observed in many papers before, but I feel the current submission makes a more focussed claim and it proposes a novel regularizer to address it in a specific manner. All in all, while the paper can certainly be improved in many ways, I feel it does enrich and advance our understanding of embeddings based methods for extreme classification,